# A Semi-Supervised Methodology for Fishing Activity Detection Using the Geometry behind the Trajectory of Multiple Vessels

**DOI:** 10.3390/s22166063

**Published:** 2022-08-13

**Authors:** Martha Dais Ferreira, Gabriel Spadon, Amilcar Soares, Stan Matwin

**Affiliations:** 1Institute for Big Data Analytics, Dalhousie University, Halifax, NS B3H 4R2, Canada; 2Department of Computer Science, Memorial University of Newfoundland, St. John’s, NL A1C 5S7, Canada; 3Institute of Computer Science, Polish Academy of Sciences, 01-248 Warsaw, Poland

**Keywords:** mobility-behavior detection, feature-augmented clustering, time-series classification, clustering analysis

## Abstract

Automatic Identification System (AIS) messages are useful for tracking vessel activity across oceans worldwide using radio links and satellite transceivers. Such data play a significant role in tracking vessel activity and mapping mobility patterns such as those found during fishing activities. Accordingly, this paper proposes a geometric-driven semi-supervised approach for fishing activity detection from AIS data. Through the proposed methodology, it is shown how to explore the information included in the messages to extract features describing the geometry of the vessel route. To this end, we leverage the unsupervised nature of cluster analysis to label the trajectory geometry, highlighting changes in the vessel’s moving pattern, which tends to indicate fishing activity. The labels obtained by the proposed unsupervised approach are used to detect fishing activities, which we approach as a time-series classification task. We propose a solution using recurrent neural networks on AIS data streams with roughly 87% of the overall *F*-score on the whole trajectories of 50 different unseen fishing vessels. Such results are accompanied by a broad benchmark study assessing the performance of different Recurrent Neural Network (RNN) architectures. In conclusion, this work contributes by proposing a thorough process that includes data preparation, labeling, data modeling, and model validation. Therefore, we present a novel solution for mobility pattern detection that relies upon unfolding the geometry observed in the trajectory.

## 1. Introduction

Navigation systems were developed to monitor vessel ships and ensure the security and safety of workers, passengers, and the environment, avoiding unforeseen events for marine vessels [1]. Most of the data used to track and monitor vessels’ movement come from the Automatic Identification System (AIS), which is essential for managing trajectories and detecting unusual events along the vessel voyage [2]. The volume of such data is remarkable, around 500–600 million messages per day [3]. Transceivers broadcast AIS messages containing the vessel identity and voyage information, including current location, length, type, speed, course, and other details [4,5]. It is crucial noticing that vessels are not required to use an AIS transceiver in many circumstances. However, monitoring vessel activity has become an international standard [6,7], which can be observed by the many commercial tools that offer solutions and data about fleets online (e.g., MarineTraffic (https://www.marinetraffic.com/ accessed 9 August 2022) and VesselFinder (https://www.vesselfinder.com/ accessed 9 August 2022)).

The volume, uncertainty, and sequential nature of the spatio-temporal data fostered research on AIS-based solutions driven to monitor [8,9] and increase safety [10,11,12], expanding our awareness about the marine traffic [13,14,15]. Along with similar premises, other authors used the AIS data to understand and forecast vessels’ trajectories [16,17] or understand their environmental impact [18,19]. Some applied their efforts to detect anomalies among AIS transmission [20,21,22,23,24] and the sub-patterns of vessel mobility [25,26,27,28]. However, few works have focused on detecting fishing activities using AIS data [29,30].

Monitoring fishing vessels present a vital role in the maritime environment because of the Illegal, Unreported, and Unregulated (IUU) fishing that jeopardizes the sustainability of the fishing industry [31,32,33], which might harm the living condition of fishing villages and communities in the surroundings. IUU fishing describes irresponsible fishing activities, such as fishing in prohibitive areas, re-flagging vessels to evade controls, and failure to report catches. To ensure sustainable fishing activities, the United Nations Convention on the Law of the Sea (UNCLOS) took the initiative to create the Monitoring, Control, and Surveillance (MCS) (https://www.un.org/Depts/los/convention_agreements/texts/unclos/unclos_e.pdf accessed 9 August 2022). The goal of MCS is to gather fishing information, conduct plans and strategies for fishing management, and ensure that adequate control measures are implemented [34]. In this context, automatic detection and identification of fishing activities are essential for enforcing the agreed policies related to fishing activities.

Many efforts were spent on searching for techniques to detect fishing activity in illegal, prohibitive, and protected areas. Most of those techniques were based on remote sensing, which through satellite images used as input, applied computer vision techniques (e.g., segmentation and classification) to track fishing vessels that avoid detection by spoofing or intentionally interrupting AIS transmissions [35,36]. Such a practice is recurrent and has been the subject of many governmental research incentives, such as the one between the USA’s Defense Innovation Unit (DIU) and the Global Fishing Watch (GFW) for detecting dark vessels, namely the xView3 IUU challenge (https://iuu.xview.us/ accessed 9 August 2022). Other approaches for detecting fishing activities use trajectory segmentation strategies to partition the trajectories into fishing or not fishing behaviors that rely on primarily kinematic features (i.e., features related to the object’s motion). These techniques are based on geographic distance error [37,38] thresholds, optimization of cost-functions [39,40], or transition state matrices [41]. These techniques are not explicitly designed for fishing detection and have limited applications.

Unlike the approaches above, de Souza et al. [30] developed three different approaches to detect and map fishing activities according to their gear. In this work, each approach was developed to identify fishing activity related to a specific gear from satellite AIS data. Although they used AIS data to assess trajectories’ spatial and temporal distribution, those approaches were designed for a particular fishing gear, which is not generalized for other conditions. Thus, in order to detect fishing activities regardless of gear type, Behivoke et al. [29] applied a random forest model to process GPS tracking of small-scale fisheries. However, such an approach is focused on a local region, which cannot be generalized because vessels present different movement patterns according to the fishing location. Neither frameworks deal with the stream AIS data, which requires fishing detection in near real-time to support the monitoring and supervision of IUU activities. Therefore, we propose an approach that relies only on the AIS messages to detect fishing activity using streams of AIS data. Thus, the proposed approach focuses on the behavior associated with the fishing activity by analyzing the geometry of the vessel trajectory.

This paper contributes a semi-supervised methodology for fishing detection using terrestrial-sourced AIS data streams and Recurrent Neural Networks (RNN) for leveraging the geometry of the vessel trajectories. In this context, labels are required to perform the fishing detection, but the AIS data have no reliable label identifying fishing activity on the AIS finer-grained message. Consequently, our semi-supervised methodology employs clustering techniques to detect patterns in vessel motion. The clusters are created over a subset of derived features that distinguish the movement patterns throughout a trajectory. Subsequently, post-processing is performed over the clustering results to label each message. Such labeled messages compose a sequential trajectory used to train and test the proposed lightweight Elman’s RNN [42] with a multivariate non-linear decoder. We provide an extensive benchmark of neural network architectures for detecting fishing activity from AIS data streams, in which our proposed network achieves about 87% of the *F*-score on trajectories of unseen vessels. Our proposed network can support the monitoring of fishing activities in real-time. Moreover, our framework can be adapted to deal with concept drift, which typically occurs in stream data. In this scenario, the movement patterns can be different due to the fishing gear, region, year, or season.

The proposed approach contributes to a timely topic of broad impact with applicability to several tasks capable of spotting fishing activities in order to assist in IUU scenarios. Such capability is of considerable concern due to the negative impact that IUU might cause on environmental and climate effects [43]. For instance, a pelagic species can be taken to extinction if no IUU enforcement policy is put into practice [44]. Therefore, our analyses were focused on fishing-intensive locations across Pacific Canada, but it has worldwide generalization capability if the required data are provided.

In order to describe our methodology and discuss our findings, this paper is organized as follows: Section 2 details the dataset and our method, where we present in Section 2.2 our unsupervised labeling part of our semi-supervised methodological approach, and Section 2.3 presents the supervised step based on time-series classification. We then describe in Section 3 the results, and Section 4 elaborates on general discussion points and concepts. Finally, the conclusions and final remarks are described in Section 5.

## 2. Materials and Methods

As unlabeled data are abundant in several application domains (e.g., text and bioinformatics), semi-supervised learning has been applied to infer labels based on fundamental properties of the data and further perform the classification tasks [45]. Such a process can be conducted by combining unsupervised and supervised models [46]. Since AIS messages lack labels of fishing activities, we propose a semi-supervised framework to support detecting fishing activities in real-time. In this scenario, the fishing activity is characterized as chaotic (i.e., abrupt) movement patterns caused by variations in the vessel’s course and speed. Therefore, two main steps are employed in our proposed frameworks: (i) an unsupervised step to label the messages, followed by (ii) a supervised step that trains a time-series classification model for the message-level fishing detection task. Figure 1 illustrates one of the contributions of our proposed methodology, in which streaming AIS data are collected to feed an RNN model. Such a model provides the message-level fishing detection task classification in near real-time.

The rest of this section is organized as follows. Section 2.1 describes our AIS dataset; Section 2.2 presents the unsupervised approach for label inference; the supervised approach to performing the time-series classification task is provided in Section 2.3.

### 2.1. Dataset

AIS messages contain information on the static and dynamic vessel trajectories’ attributes, with the Maritime Mobile Service Identity (MMSI) commonly used as the vessel identifier. This paper uses the MMSI as the vessel identifier, accompanied by the latitude, longitude, Speed Over Ground (SOG), and Course Over Ground (COG). As our framework focus on fishing detection, the AIS data used in the experiments are of fishing vessels from the Strait of Juan de Fucain April, May, and June 2020. These trajectories are publicly available at https://marinecadastre.gov/AIS/ (accessed 9 August 2022) and were collected using terrestrial receivers covering Canada and United States.

The dataset was preprocessed by removing invalid and duplicated messages and rounding latitude and longitude to the fourth decimal place (10 m of resolution). We consider messages with a negative SOG value or COG value outside the range [0,360] to be invalid. Messages with SOG ≤0.5 were also removed as they indicate no mobility pattern for the analysis. Figure 2 shows the fishing vessel trajectories used for the experimentation on the left and the edge-bundling visualization underlining the traffic intensity on the right.

### 2.2. Unsupervised Step

Our methodology starts with the cluster-based approach to label AIS messages as sailing and fishing, which is described in Figure 3. Typically, the movements of fishing activities differ from the movements established in the pathways (i.e., shipping lanes) and in-line sailing (Step 1 in Figure 3), which are typically followed by vessels such as cargo and tankers [47]. Based on these characteristics, features are extracted to capture the differences in speed and course, commonly picturing sharp curves when in a *Mercator Projection* (Steps 2 and 3 in Figure 3). Such curves and other movement patterns are related to activities beyond regular sailing and are commonly observed in fishing vessels. Next, the *k*-means [48] algorithm is applied to these features to assign clusters to the input data, resulting in multiple clusters that represent different movement patterns (Step 4 in Figure 3). Later in the unsupervised pipeline, post-processing is performed to label highly populated clusters as sailing while aggregating the remaining clusters to be labeled as fishing.

In order to capture the sharp curves and strait rides, the features are extracted by sharing the trajectory information between consecutive AIS messages using the Moving Average and Sum. Initially, the difference between adjacent messages is computed over SOG and COG attributes, resulting in acceleration and rate of COG (RCOG) features. The difference between an attribute of two consecutive messages is defined as:(1)$$\begin{array}{c} d_{t}^{a}=m_{t+1}^{a}-m_{t}^{a} \end{array}$$
in which $d_{t}^{a}$ represents the difference of attribute *a* at time *t* (which can be negative), and $m_{t}^{a}$ corresponds to an attribute *a*, such as SOG and COG, associated with a message *m* at time *t*. In the case of RCOG, the smaller signed angle between the difference of COG is computed because we assumed that the vessel would take the short path to perform the turn. In this case, the vessel is able to turn 180 degrees to the right (positive angle) or 180 degrees to the left (negative angle). This calculation is defined as follows:(2)$$\begin{array}{c} {\mathrm{RCOG}}_{t}=\left\{\begin{array}{c} d_{t}^{\mathrm{COG}}-360\;\mathrm{if}\;d_{t}^{\mathrm{COG}}>180\\ d_{t}^{\mathrm{COG}}+360\;\mathrm{if}\;d_{t}^{\mathrm{COG}}<-180 \end{array}\right. \end{array}$$
in which $d_{t}^{\mathrm{COG}}$ represents the difference of COG at time *t* as defined by Equation (Equation 1), and ${\mathrm{RCOG}}_{t}$ corresponds to the smaller signed angle. Figure 4 illustrates the vessel turning, in which the red curve shows the smaller angle that is calculated. Moving Average (MA) and Moving Sum (MS) are computed to extract spatial and temporal dependencies of a local segment of the trajectory. This process smooths the data by keeping the information shared between consecutive AIS messages. MA is performed over acceleration to capture the speed average in a local segment. On the other hand, MS is computed over the RCOG to obtain the total curvature present in that segment. The window size hyperparameter of the MA and MS defines the number of consecutive AIS messages. Such hyperparameter was defined using a fixed (A) number of messages, (B) time-window within AIS messages (in minutes), and (C) distance-window between messages (in meters). We conducted a visual analysis of the sliding window size to assess the impact of approaches (A), (B), and (C).

As the broadcast interval between consecutive messages varies greatly, these designed approaches tackle the problem with a distinct perspective of the messages’ interactions. Even with a predefined course, the trajectory is determined based on local decisions during the vessel voyage, indicating a low temporal dependency between messages. This short-time dependency motivates our assumption that each feature-set will be relevant to detecting fishing activity from different perspectives. Therefore, by assessing different ways of slicing the data, it is possible to capture different spatio-temporal granularity within the messages, tending to aid the detection activity.

The extracted features are clustered with *k*-means to separate them into movement patterns. We observed that the higher the number of clusters, the more movement patterns are captured during the clustering. Such an observation comes from the hierarchical nature of vessels’ movement, including larger patterns composed of smaller ones. However, when increasing its complexity by capturing more labels, no improvement is observed for the annotation of fishing activities because it starts to capture subtle movement patterns within the pathways. Therefore, the number of clusters was selected based on a tradeoff between the Davies–Bouldin index (DBI) and visual analysis of the clustering results. Thus, the lower the DBI index, the better the clustering, meaning less overlap between clusters.

Post-processing was performed to annotate the messages into sailing and fishing. Thus, sailing messages were associated with the highly populated cluster, while fishing messages correspond to the remaining ones. Subsequently, we reclassified them to remove noisy elements, which are messages that are not entirely similar to adjacent ones in the trajectory sequence. Figure 5 illustrates the noise messages by a red circle, in which a set of adjacent messages classified as fishing will be reclassified as sailing. Such a reclassification requires a predefined minimum number of messages as the threshold. Thus, the label is changed to the opposite class whenever a set of adjacent messages fails to meet the threshold. Ultimately, each message in the vessel trajectory is associated with one label.

### 2.3. Supervised Step

By having the labels assigned to each AIS message in the dataset, our methodology follows by building a classifier based on the continuous-valued attributes of the messages that can distinguish when the vessel is sailing or fishing (see Figure 6). Because the AIS messages behave similarly to a multivariate time series, the model was built to consider variable-and temporal-related information from multiple trajectories. Therefore, our model will learn from the continuous-valued features in the message by observing how they unfold through time. Simultaneously, the model will learn from multiple different trajectories from where it will capture its shared movement patterns.

In this scenario, the employed dataset $X\in R^{m\times \tilde{w}\times v}$ has *m* different trajectories, each one with a varying length and represented as $\tilde{w}$. This way, a given trajectory *i* at a given time *j* has $X_{ij}=\left\{latitude,\;longitude,\;COG,\;SOG\right\}$ as attributes. It is worth mentioning that our approach assumes that the time between consecutive AIS messages is ordered as the messages were captured by the receiver (see Figure 1). However, the time between messages of the same trajectory varies greatly from minutes to hours, increasing uncertainty in understanding the vessel’s movement patterns [49]. Nevertheless, vessels usually do not fish far from the shore, tending to be in the range of terrestrial AIS receivers. This behavior increases the granularity of the messages significantly and smooths the timing irregularity naturally. Regardless, we assume such a difference in time as noise inherent to the data domain. Therefore, the model was trained with enough data to avoid substantial time variations that usually affect the models’ inference performance and usability.

We reserved 50 trajectories of unique vessels based on their MMSI for the final test, 15 for hyperparameter tuning and validation, and the remaining 535 for the training. The attributes of the training data went through min-max and *z*-score normalization, and the validation and test data were normalized with their normalization parameters. Using a prefixed window of size *w* aided by the sliding window algorithm, the data were batched on the temporal axis, so all batched samples have the same temporal length. It is worth mentioning that the messages that do not fit a complete mini-batch of window-size *w* are discarded. The dataset $X\in R^{m\times \tilde{w}\times v}$ now becomes $X\in R^{b\times w\times v}$, where *b* is the total number of batches considering all different trajectories with a window of size *w*, i.e., number of AIS messages in sequence, having b≫m. Such data were then used to feed our recurrent layer:(3)$$\begin{array}{c} h_{t}^{\left[1\right]}=\tanh\left(\left(W_{ih}^{\left[1\right]}\cdot \delta \left(x_{t}\right)+b_{ih}^{\left[1\right]}\right)+\left(W_{hh}^{\left[1\right]}\cdot h_{\left(t-1\right)}^{\left[1\right]}+b_{hh}^{\left[1\right]}\right)\right)\\ h_{t}^{\left[2\right]}=\tanh\left(\left(W_{ih}^{\left[2\right]}\cdot \delta \left(h_{\left(t-1\right)}^{\left[1\right]}\right)+b_{ih}^{\left[2\right]}\right)+\left(W_{hh}^{\left[2\right]}\cdot h_{\left(t-1\right)}^{\left[2\right]}+b_{hh}^{\left[2\right]}\right)\right)\\ h_{t}^{\left[3\right]}=\tanh\left(\left(W_{ih}^{\left[3\right]}\cdot h_{t}^{\left[2\right]}+b_{ih}^{\left[3\right]}\right)+\left(W_{hh}^{\left[3\right]}\cdot h_{\left(t-1\right)}^{\left[3\right]}+b_{hh}^{\left[3\right]}\right)\right) \end{array}$$
where $W_{ih}^{\left[1\right]}\in R^{w\times s}$ and $W_{ih}^{\left[2\right]},W_{ih}^{\left[3\right]},W_{hh}^{\left[1:3\right]}\in R^{s\times s}$ are the weights, $b_{ih}^{\left[1:3\right]},b_{hh}^{\left[1:3\right]}\in R^{s}$ is the bias; $h_{t}^{\left[1:3\right]}$ is the hidden state vector of different recurrent layer cells stacked vertically and numbered orderly from 1 to 3; and δ is a Bernoulli dropout function. Such architecture block refers to a triple-layer Elman’s Recurrent Neural Network (RNN) [42], also known as *simple* or *vanilla* RNN, in which each $x_{t}\in X$ is a mini-batch of the dataset at instant *t*, *s* denotes the size of the hidden layers (i.e., latent-variables dimension), and the last hidden state $h_{t}^{\left[3\right]}$ is the input of a multi-task, -variate, and -layer perceptron (MLP) [50] decoder.

The decoding of the latent variables from the last RNN cell is usually performed by one or a few linear-dense layers. However, considering that our task is based on multivariate temporal data for predicting a binary output shared across variables, our MLP decoder first encodes the data on the variable axis to later decode in sequence with the temporal axis, deriving the *logits* that will become the labels after the activation function is applied:(4)$$\begin{array}{cc} \hat{y} & =\sigma \left(W_{w}\cdot \mathrm{ReLU}\left(W_{v}\cdot \left(W_{h}\cdot h_{t}^{\left[3\right]}+b_{h}\right)\right)\right) \end{array}$$
where $W_{h}\in R^{v\times s}$, $W_{v}\in R^{s}$, and $W_{w}\in R^{s}$ are the weights and $b_{h}\in R^{s}$ the bias. The decoder operates by translating the latent variables from the temporal encoder into the target labels. In such a way, through $W_{h}$ and $b_{h}$, the model further transforms the multiple variables from the AIS message into further hidden weights, resulting in $h_{t}\in R^{b\times h\times h}$. The results are then transformed by $W_{v}$ and summarized into two dimensions, generating a tensor $h_{t}\in R^{b\times h}$. Such a tensor goes through a Rectified Linear Unit (ReLU) activation, bringing non-linearity to the pipeline and allowing further performance improvement in the decoding process. Then, the activated output goes through $W_{w}$, where it will achieve $h_{t}\in R^{b}$ (the *logits*), and after a sigmoid activation and a rounding function, will be in the binary scale. It is noteworthy that the decoder leverages from a single bias term (i.e., $b_{h}$) due to the reduction in the dimension of the temporal-and variable-axis, which would result in a bias term of a single parameter, which does not add much to the learning process.

The model was trained using a combination of the Center Loss (CL) [51] and the Binary Cross Entropy (BCE) [52] on a 75/15 weighting ratio, respectively. The optimizer used was AdamW [53], corresponding to Adam with Decoupled Weight Decay Regularization. The training process used a learning rate of $10^{-3}$ and gradient clipping on the max grad-norm with 1.0 as the threshold. The learning rate employed in these experiments was scheduled to reduce whenever the model encountered an improvement plateau during the training process. The network was trained in looping until no improvement was observed in the BCE loss on the 15 vessel trajectories reserved as validation. However, the final test was conducted on the 50 vessel trajectories of unseen and untrained data using the model trained up to the epoch where the lowest BCE value was observed on the 15-vessel dataset.

Notice that we do not use the CL to assess the validation on the 15-vessel dataset because the CL is a parameter-dependent loss function optimized concurrently to the model. Along these lines, we explored the impact of different hyperparameters, such as the hidden size of the encoding operations and the window size of the temporal batching step. For the learning of the fishing forecasting task, we evaluated how they affected the performance of different RNNs, such as Elman’s RNN, Gated Recurrent Unit (GRU) [54], and Long-Short Term Memory (LSTM) [55]. Finally, such extensive experiments were used to set a broad benchmark of unsupervised semantics for fishing activity detection to guide future research with similar premises and related tasks.

We explored the results using a set of broadly known evaluation metrics, namely Precision, Recall, and F-score (i.e., F1) [56]. Precision depicts the model’s specificity by assessing predictions’ True Negative Rate (TNR). The recall (or sensitivity) assesses the Positive Predictive Value (PPV). F1, on the other hand, is the harmonic mean of both precision and recall. Such evaluation metrics are defined as follows:(5)$$\begin{array}{c} \mathrm{Precision}=\frac{TP}{TP+FP},\;\;\mathrm{Recall}=\frac{TP}{TP+FN},\;\mathrm{and}\;\;F1=2\times \frac{\mathrm{Precision}\cdot \mathrm{Recall}}{\mathrm{Precision}+\mathrm{Recall}} \end{array}$$
where TP refers to the True Positive Rate, FP to the False Positive Rate, and FN to the False Negative rate. Along with the corresponding experiments, we evaluated each dataset class separately with the three metrics. Additionally, we included the unweighted average of all classes, which refers to a pessimistic view of the model in the face of heavily unbalanced datasets, such as in the case of fishing detection that has majority sailing labels rather than fishing labels. As the models tend to perform better in the majority class rather than the opposed one, the unweighted average will bring the overall performance down closer to the minority label performance. These experiments described in Section 3 were carried out on a Linux-based system with 40 CPUs, 128 GB of RAM, and a GeForce RTX A100.

## 3. Results

### 3.1. Analysis of the Unsupervised Approach

As previously mentioned in Section 2.2, during the unsupervised pipeline, our approach employed *k*-means to label the AIS messages of the fishing vessels using two attributes from the messages as features: (1) MA of the acceleration and (2) MS of the RCOG. Three techniques were applied to define the sliding window size: (A) traditional approach based on a prefixed number of consecutive messages (message-based), (B) a time interval among consecutive messages that share information (time-based), and (C) distance range up to where the messages should be considered (distance-based). For this approach, the number of clusters defined for the *k*-means algorithm was evaluated based on DBI, while a visual analysis of the sliding window size was conducted to assess the impact of the three approaches. We explored a post-processing approach of inverting the labels when less than five messages were mapped consecutively into the same class.

#### 3.1.1. *k*-Means Clustering Analysis

To analyze the number of clusters, we fixed the sliding window size as (A) 10 messages, (B) 10 min, and (C) 5000 m. These values were selected by assuming that the average time between AIS messages is around 1 min and 500 m. Therefore, 10 AIS messages correspond to approximately a 10-min time window and a 5000-m distance window. The DBI values obtained for the three approaches are illustrated in Figure 7. It is possible to observe that when the number of clusters increases, the DBI value increases, indicating an increase in the overlap between clusters. However, DBI values decrease before stabilizing when using the distance-window technique, indicating that this approach requires more clusters to reduce the overlapping. In addition, such an approach provides less overlapping among clusters than the other two techniques. The mentioned overlapping can be observed in a visual analysis, where some messages in shipping lanes were classified as fishing activity. Figure 8 illustrates the mislabeled events, where the red circle shows messages classified as fishing activity in well-known shipping lanes [57].

This misclassification occurs because lanes can have smooth curves captured by the *k*-means algorithm when increasing the number of clusters. However, based on the visual analysis, the increase in the number of clusters provided a better representation of the sharp curve and speed variation. This behavior is illustrated in Figure 9, where the sharp curves are better represented with a higher number of clusters. According to the visual analysis and DBI measure, eight clusters for message-based and time-based approaches revealed a more suitable arrangement capable of capturing sharp curves and speed variations when vessels sail outside the shipping lanes. In the distance-based approach, 12 clusters are the most suitable value to capture the movement patterns that characterize fishing activities.

#### 3.1.2. Window-Length Analysis

The DBI values were similar when adjusting the window size regardless of the criteria used in the AIS information-sharing processes. Therefore, the number of clusters were kept as the obtained values in the analysis on Section 3.1.1 to select an adequate window size. Such a selection was based on the visual analysis of the trajectory geometry. Assuming that the average time between AIS messages is around 1 min, a 10-min time window contains about 10 AIS messages. However, as the time between AIS messages is not fixed (see Section 2.3), the time-based window size provides a varying number of AIS messages when applying the Moving Average and Sum in the information-sharing stage. This flexibility allows the windowing to capture more subtle patterns and ignore messages with significant temporal gaps, reducing the uncertainty that lies with AIS message transmissions.

Additionally, the same scenario can be pictured in the distance-based approach, in which, assuming that the average distance between AIS messages is around 500 m, a 5000 m distance window contains about 10 AIS messages. This approach criteria will not provide fixed-size windows in terms of AIS messages, similarly to the time-window approach, because the speed changes with voyage duration. Figure 10 illustrates that the time-based criteria capture finer curves, while the message-based ones could not capture the same curves when using a fixed number of 10 messages. It also illustrates that the distance-based criteria are more robust in terms of overlapping (i.e., shipping lanes were not classified as fishing activity). However, the distance-based window fails to detect some fishing patterns in the trajectory, which is not preferable for our task. This behavior is illustrated in Figure 11, where red circles show the patterns that were not detected.

It is important to note that using too wide window sizes during the information-sharing phase can jeopardize the semantic extraction from the vessel trajectory. For instance, the chaotic or abrupt movement patterns considered in fishing activities in our scenario are related to changes in the vessel’s course and speed. These patterns might be lost if a single class dominates the windowed sequence. However, using a larger window reduces the mislabeling on top of shipping lanes as those small changes in the course will not be considered abrupt in the in-lane sailing context. As a matter of exemplification, Figure 12 illustrates the trajectory labeling using 7, 10 and 15 min for the time-based windowing. Such an image reveals that time-based windowing with 10 min captures smoother movements while using 15 reduces the mislabeling and does not detect some patterns. Therefore, after visual analysis, 10 messages, 10 min, and 5000 m seem to be adequate hyperparameters for annotating fishing activities in our dataset.

#### 3.1.3. Fishing Detection Feature Analysis

In order to compare the features produced by observation, time, and distance-based approaches, we used a window size of 10 messages, 10 min, and 5000 m, which were defined by the previous analysis in Section 3.1.2. Figure 13 illustrates the features colored by the labels from the *k*-means algorithm using the number of clusters defined based on the analysis presented in Section 3.1.1. Thus, 8 clusters were used for observation-window and time-window techniques, while 12 clusters were applied for distance-window. Using the time-based information-sharing approach, the image indicates that the acceleration-related feature becomes scattered for low RCOG values. Contrarily, the RCOG becomes more scattered on low acceleration values when using a fixed number of AIS messages. On the other hand, the distance-based approach increased the scatteredness of both RCOG and acceleration-related features, leading to less overlapping as observed with DBI values.

Accordingly, RCOG values seem to be the most relevant feature in detecting chaotic-like patterns. However, the acceleration prevented noise among the more minor labeled sequences, avoiding spotting an incorrect sailing-related message between fishing-related messages. Considering that correctly labeling the dataset with fewer uncertainties and noise is the goal of our methodology, we concluded that the acceleration is key-feature for the fishing detection task, leading to a more precise pattern detection. Additionally, Figure 14 shows the features colored after the post-processing, in which it is noticeable that sailing activities tend to be in the same regions, while fishing activities are detected in messages that show a diverging pattern. This behavior shows that messages in a shipping lane tend to keep the same acceleration and direction since this region is close to zero, indicating a slight variation in the movement pattern.

### 3.2. Analysis of the Supervised Approach

In the supervised context, we proposed a neural network architecture solution (see Section 2.3) capable of detecting fishing and sailing patterns in AIS streams based on our unsupervised methodology (see Section 2.2). Due to the sequential nature of AIS messages, we chose to use Recurrent Neural Networks (RNNs) in a time series classification task. In this sense, Table 1 presents the results of the benchmarks performed on our architecture subject to different *window*-and *hidden-layers* sizes. The metrics are unweighted, and the *support* stands for the number of instances of a given class from which the respective metrics were calculated. Among them, we vary the size of the training’s sliding window, which indicates the number of consecutive AIS messages digested by the model that increase or reduce dependency and the size of the hidden layers, which limits the neural network’s learning power by controlling the dimensions of the hidden weights. Both hyperparameters directly interfere with the number of internal parameters of the model, being the hidden layers’ size that majorly affects the model complexity.

Table 1 reveals that the observation-based features are the ones that yield the best performance among the two other labeling techniques. This means that the model can better identify the labels of the fixed-size observation-based features using the raw AIS message from the message stream. The second-placed is the time-based pipeline, and the third is the distance-based one. However, we noticed that we cannot assess these models in terms of accurate fishing detection straightforwardly. Recall that models were created based on an unsupervised methodology without label verification, besides the visual analysis of the trajectories and cluster assignments. Therefore, as there is no common ground for comparison, it is infeasible to determine which of these techniques is more accurate.

Nevertheless, we assessed other features that could better describe the problem being solved, such as the low temporal dependency between messages. That improvement is due to the superior performance of 10 over both 5 and 15 messages as the window size. This means that there is no need to look overly far in the AIS message sequence to understand the movement patterns of fishing vessels, as those are based on the local movement.

Due to the low temporal dependency inherent to the problem domain, we used an Elman’s RNN cell as part of our architecture instead of a GRU or LSTM. Elman’s are one of the first RNNs proposed in the literature, and it has a simple recurrent mechanism that leverages few hidden matrices and non-linearities compared to GRUs and LSTMs. That makes Elman capable of capturing recurrent patterns in the near past using varying-size sequences. Because long-term patterns do not significantly improve the process, the related long-term learning mechanisms present in GRUs and LSTMs would be underused. For such cases, having cells with such long-term modeling capabilities makes the problem more challenging. That challenge is due to the complexity and increased number of parameters of those cells without a significant gain in performance that would motivate their use as part of the architecture. In other words, we decided to leverage an “almost-good” lightweight model due to its low complexity and training time over an alternative unit.

Table 2 shows the results for the GRU and LSTM units using a fixed window of size 10 and a hidden layer size of 64, the same used by our architecture from now on. We can observe that the LSTM and GRU results are slightly better than Elman’s RNN results in Table 1. However, the number of parameters required by each alternate architecture is alarmingly higher. Using such a network in a streaming setting would require more training time to fine-tune the network with daily-updated data, and the same would hold for the inference time. Therefore, regardless of the set of features used for the detection task, our architecture’s number of internal parameters using Elman’s RNN is in the other of 20,000, a fraction of GRUs and LSTMs that have, respectively, about 60,000 and 80,000 parameters.

As a consequence of the lightweight capability of our neural network and the previously mentioned difficulty in assessing the performance of the labeling techniques in a common ground, in our final experiment, we proposed a solution by merging the three techniques into a unique neural ensemble in the form of a voting classifier. Such an approach is based either on counting the frequency of labels and going with the majority (i.e., soft voting) or averaging the probabilities from each model before applying the sigmoid activation and then a rounding function (i.e., hard voting) to jointly-built label.

This merging approach achieved a reasonably conservative model shared across all the semantic feature extraction techniques we proposed during our unsupervised pipeline. This means the model tends to misclassify a pattern favoring the sailing class more frequently, reducing false alarms but increasing unspotted fishing. That behavior can be observed in both soft and hard voting results in Figure 15, where the model achieved 99% of True Negatives (TN), 62% on the True Positive (TP), 1% of False Negatives (FN), and 38% of False Positive (FP). In both soft and hard voting cases, the output model has three times the number of parameters it initially had, so instead of 20,000 parameters, it now has about 60,000, which are shared among three neural networks, each with one triple-layered Elman’s RNN. The number of parameters is smaller than a single triple-layer LSTM and almost equal to a triple-layer GRU. Because of that, using a GRU or an LSTM unit could scale the number of parameters to a point where the model’s streaming capability would be jeopardized due to the time required to train, re-train, and infer over the ensemble. The tradeoff of parameters and performance favors our proposal, which showed to be computationally less complex and equally efficient across the three unsupervised labels.

## 4. Discussion

The main challenge in label annotation using unsupervised approaches is that the obtained labels are not verified, i.e., they are an estimate of probable activity occurrences. In our scenario, fishing activity is characterized as abrupt changes in the vessels’ motion, which are detected by analyzing the variations in their course and speed. Consequently, our approach labeled messages in shipping lanes as fishing activity due to the trajectory’s smooth curves and slight speed variation. On the other hand, the multiple movement patterns within fishing labels (see Section 2.2) tend to be related to how fishers deploy and collect fishing gear during fishing activities of different kinds. Nevertheless, details of the patterns meaning are out of the scope of the proposed work due to difficulty validating the unsupervised approach.

In a different perspective, where fishing is prohibited, such as in protected marine areas, having labels for fishing detection tasks is infeasible because whoever engages in IUU activity covers up their tracks once they knowingly commit an environmental conservation crime. Even if fishing activity data in allowed areas are available to label the dataset, the model still needs to detect minor fishing pattern changes, which would come from the fishers’ tentative in hiding their intentions when fishing in protected marine areas. In such a case, even with a labeled dataset, it would not be possible to estimate the efficacy of our technique in the face of an IUU activity. However, given the shared nature of the IUU fishing activity, the fishing patterns should behave similarly. Therefore, although we cannot verify the model in the face of such constraints, having information about potential fisheries in prohibitive areas can bring further information to the public policies on sustainable fisheries planning and open new doors for further research on similar lines where the required data are available.

An alternative to such problems would be to have an expert as part of the model (see Figure 16), which mitigates the need for more data, as the expert would be able to validate the movement patterns further. This validation would serve as a delayed input of the network model through transfer, incremental, and active learning. Thus, the training process is repeated to accomplish a refined model for the fishing task as more reliable data are received. In this case, it is essential to notice that, due to the reduced complexity of our proposed neural network model, the model’s capability to learn more intricate patterns arising from the data is limited. Suppose the expert observes relevant patterns beyond those we see through our unsupervised methodology, then a more robust neural network architecture can potentially learn other relationships, such as in long-term dependencies. However, our data support that the changes in mobility patterns are better captured when looking at smaller temporal windows tied to short-term temporal dependency behavior.

## 5. Conclusions

This paper proposed a semi-supervised methodology for fishing detection on the AIS-message level. Our contributions include a thorough label annotation methodology for fishing activity based on clustering semantic features and detecting fishing activity using a lightweight neural network. Specifically, our unsupervised contribution stands on the chaotic behavior detection, which leads to abrupt geometric changes in the movement patterns of vessels that correlate to how fishers deploy and collect fishing gear during fishing activities of different kinds. On the other hand, the supervised contribution contains a benchmark of neural network architectures, in which we introduced a simple neural network model with comparable performance. Our developed model is based on multi-layer Elman’s RNNs with a non-linear multi-task and multivariate MLP, containing a fraction of the number of parameters of more recent options, such as the GRUs and LSTMs recurrent units.

The unsupervised analysis of results revealed that the temporal dependency of AIS messages when tracking changes in movement patterns is short, meaning that more recent messages are more important than far-away past messages in the sequence for this task. We explored such characteristics using different approaches to *share information* across consecutive messages in a trajectory. More specifically, we applied the Moving Average and the Moving Sum to the vessels’ course and speed information using different ways to select the sliding window size. We proceed using a fixed number of messages, a time range in minutes considering all trajectory-related AIS messages transmitted in between, and a distance range with all messages between two given locations. As part of our results, we present an analysis of each strategy, indicating where one approach is recommended over the others. Such analysis suggested that using a fixed number of messages reduces the false positive rate in the labeling when vessels are in shipping lanes where no fishing happens due to the heavy vessel traffic. However, such an approach fails to detect the subtle changes that may be a cue for the beginning of fishing activity. On the other hand, the time-based strategy captures these subtle changes in the trajectories. This strategy increases the false positive rate by labeling messages in shipping lanes as fishing activity. Lastly, the distance-based approach reduces the overlapping between groups but fails to detect abrupt changes in the vessel motion. That was due to too-long sequences of AIS messages in a distance range, which led the clusters to include mobility patterns with mixed semantics.

In the supervised analysis, we observed the difference in performance arising from the data generated by message- and window-based approaches. Even if the exact model trained on different data does not deliver comparable results, we could still notice that the model’s performance on the observation-based analysis is higher than the time-based analysis. Such difference is usually given in higher precision and recall in the sailing class for the observation-based approach. On the other hand, when it comes to the time-based strategy, we observed the opposite behavior, where we have higher precision and recall in the fishing class and lower ones for the sailing class. As the trajectories are heavily unbalanced in favor of the sailing label, the time-based approach showed less bias. Therefore, to leverage the potential of the three windowing strategies, we proposed a voting option that joins all three to build a collective conservative lightweight model.

In future works, our methodology can be explored to analyze the expansion of mobility patterns over time, potentially to assess the different intensity of vessel traffic and fishing activity spotted nearby marine protected areas. We intend to mature the supervised pipeline by adding additional features, such as the salinity of the water, seabed type, seasonal events, and other characteristics that could be related to the fishing activity in a specific area. Another possibility would be adapting the network architecture to leverage an embedding of vessel trajectories, aiming for increased performance due to the shared information in the compressed latent variable space. Our approach contributes to a more refined understanding of the mobility patterns within fishing vessels. We consider that it can be extended to further applications, such as using other vessel types or being the motivation behind analyzing other tasks from different transportation means.

## Figures and Tables

**Figure 1 sensors-22-06063-f001:**
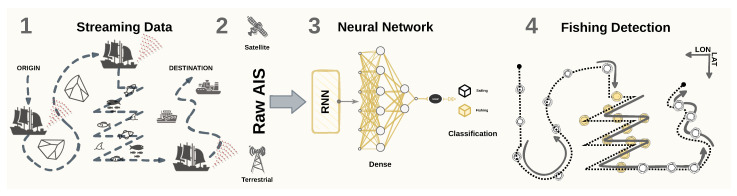
Methodological workflow for the end-to-end fishing detection task. In the diagram, (1) the data are streamed from the vessels’ AIS transceiver; (2) the AIS messages can be captured by terrestrial or satellite receivers; (3) the continuous-valued data in the AIS message are fed in raw and in sequence to the RNN-based model; (4) the model provides the decision on the vessel activity.

**Figure 2 sensors-22-06063-f002:**
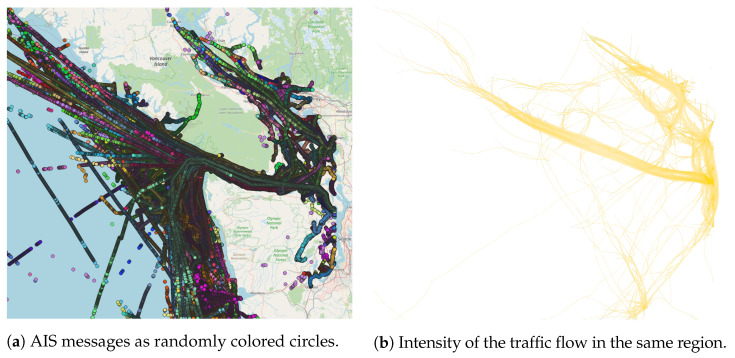
AIS messages from April, May, and June 2020 concerning fishing vessel traffic in the *Strait of Juan de Fuca*. In the image (**a**), the messages are represented as randomly colored circles, and, in the image (**b**), as a kernel-based edge-bundling visualization of traffic flows.

**Figure 3 sensors-22-06063-f003:**
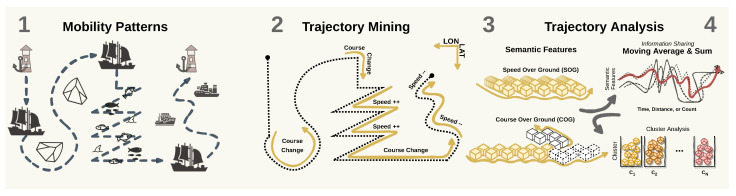
Step 1—Unsupervised workflow for the fishing detection task. According to the diagram, we follow by (1) gathering AIS data of fishing vessels; (2) capturing semantic features through a trajectory mining approach using course- and speed-related attributes; (3) sharing the trajectory information between consecutive AIS messages using the Moving Average and Sum; (4) setting the labels of the AIS messages as the cluster assignment after post-processing the results.

**Figure 4 sensors-22-06063-f004:**
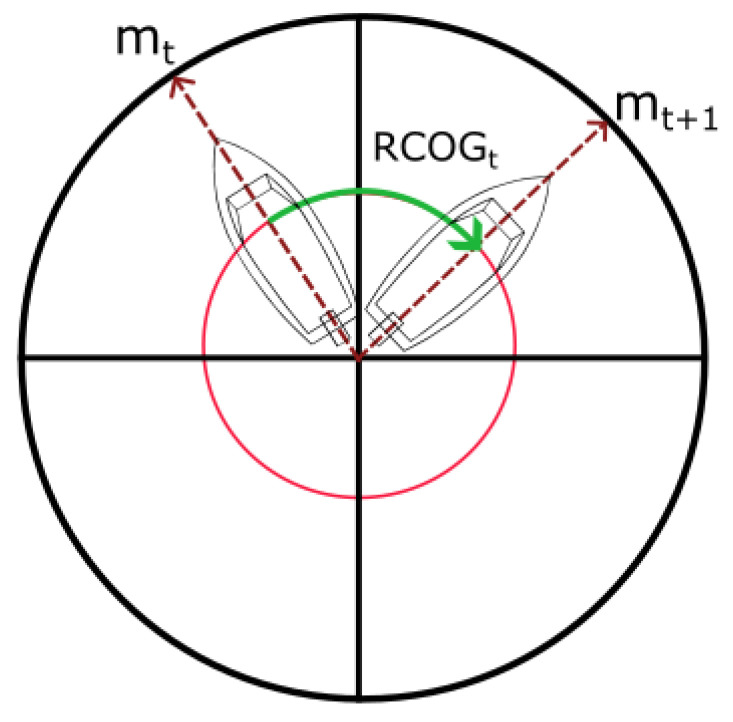
Vessel turning to the right, in which $m_{t}$ represents the COG at time *t*, the red curve represents the largest angle, and the green curve shows the smaller angle that could be followed to perform the turn.

**Figure 5 sensors-22-06063-f005:**
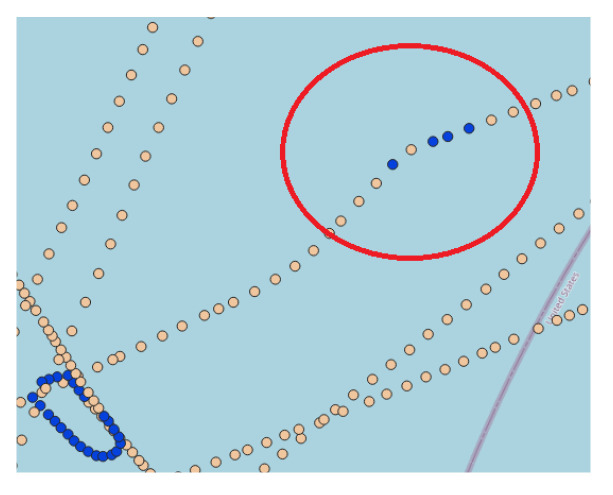
AIS messages of a vessel trajectory before the post-processing. The image highlights in red a set of messages classified as a fishing activity but refers to a slight change in course.

**Figure 6 sensors-22-06063-f006:**
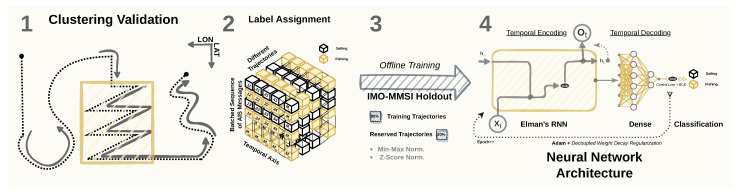
Step 2—Supervised workflow for the fishing detection task. In the diagram, (1)we use the semantic features to detect the patterns of interest in the dataset; (2) the patterns turn into labels, which are broadcasted to the messages within a vessel trajectory; (3) from the labeled trajectories, we reserve whole trajectories from unique vessels for hyperparameter tuning and the final validation test; and, (4) the non-reserved data are then normalized and fed to the neural network, which will provide us with one label for each sequence of the vessel trajectory given as input, indicating if the vessel was either sailing or fishing during the time between the last two consecutive AIS messages.

**Figure 7 sensors-22-06063-f007:**
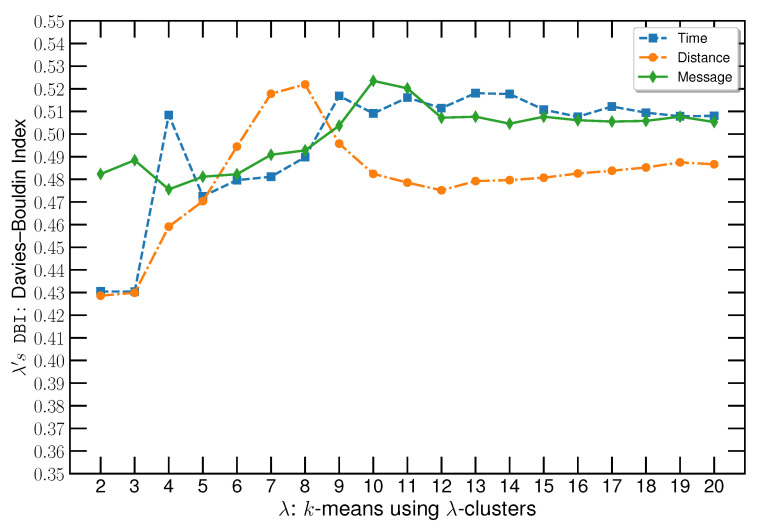
Illustration of DBI values over various clusters using different information-sharing techniques. The blue line corresponds to the results using 10 min as a fixed time, the red one to the results using 10 m, and the orange line represents the results using 10 fixed messages.

**Figure 8 sensors-22-06063-f008:**
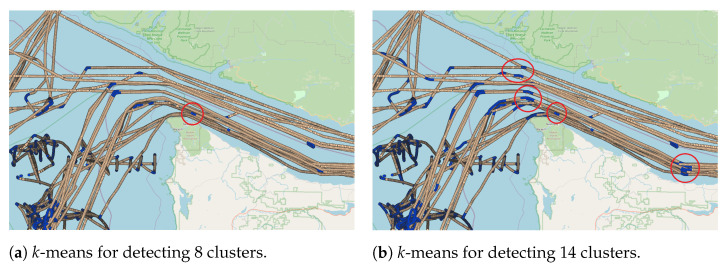
Illustration of the labeling using the fixed-number of message criteria with a value of 10 messages, in which blue dots represent the AIS messages where the fishing activity was spotted, and the red circles indicate those messages that were mislabeled. In detail, image (**a**) shows when *k*-means was applied to identify 8 clusters, while image (**b**) presents the detection results of 14 clusters.

**Figure 9 sensors-22-06063-f009:**
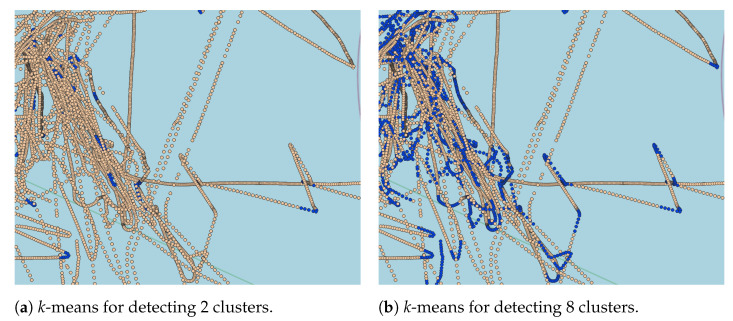
Illustration of the labeling using a fixed-number of 10 messages, in which blue dots denote the AIS messages where fishing was detected. In detail, image (**a**) shows when *k*-means was applied to detect 2 clusters, while image (**b**) shows the case when it was used to detect 8 clusters.

**Figure 10 sensors-22-06063-f010:**
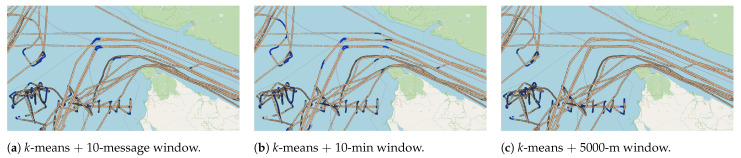
Illustrationof the labeling results using the *k*-means algorithm with 8 clusters. In finer detail, (**a**) shows the message-based windowing criteria for 10 fixed AIS messages, (**b**) presents the results using the time-based criteria with a 10 min window, and (**c**) presents the results using the distance-based criteria with a 5000 m window.

**Figure 11 sensors-22-06063-f011:**
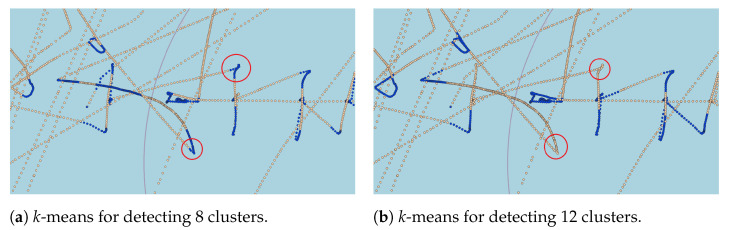
Illustration of the labeling using the time-window and distance-window criteria, in which blue dots represent the AIS messages where the fishing activity was spotted, and the red circles indicate those mislabeled messages. In detail, (**a**) shows the time-window criteria with a value of 10 min for 8 clusters, and (**b**) presents a 5000-m window for 12 clusters.

**Figure 12 sensors-22-06063-f012:**
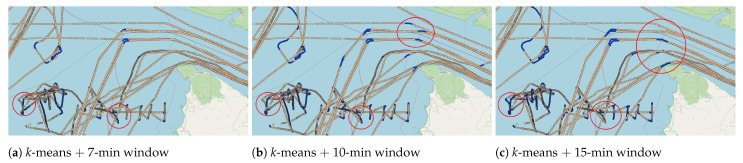
Illustration of the labeling results using *k*-means with 8 clusters, with red circles highlighting the areas of interest. In detail, (**a**) shows time-based size window of 7 min, (**b**) presents the results using 10 min, and (**c**) illustrates for 15 min.

**Figure 13 sensors-22-06063-f013:**
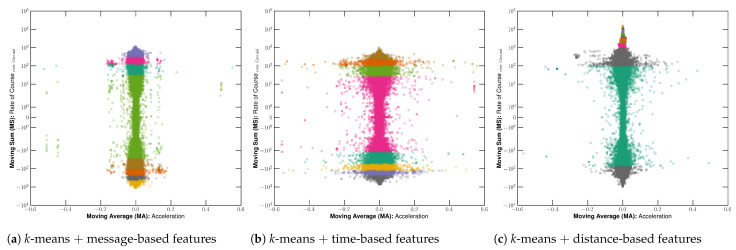
Illustration of the semantic features color-coded by the results from the *k*-means clustering algorithm before post-processing the labels, in which the features colored in blue represent potential fishing, and the others represent sailing. In finer detail, image (**a**) shows message-based features with 10 fixed observations, image (**b**) presents time-based features on a 10 min window, and image (**c**) presents distance-based features on a 5000 m window.

**Figure 14 sensors-22-06063-f014:**
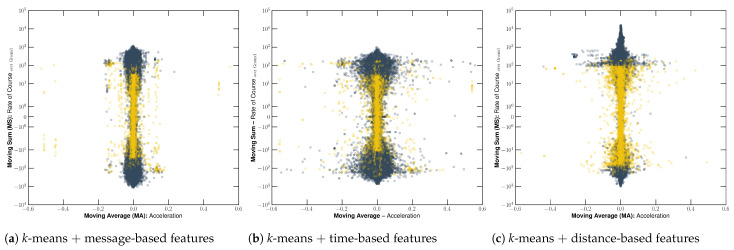
Illustration of the semantic features color-coded by the results from the *k*-means clustering algorithm after post-processing the labels, in which the features colored in blue represent potential fishing, and the others represent sailing. In finer detail, image (**a**) shows message-based features with 10 fixed observations, image (**b**) presents time-based features on a 10 min window, and image (**c**) presents distance-based features on a 5000 m window.

**Figure 15 sensors-22-06063-f015:**
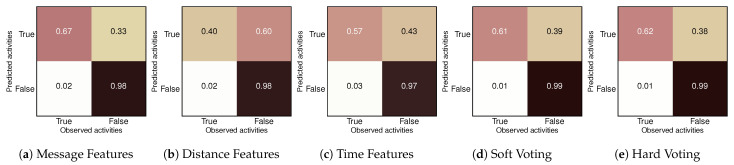
Confusion matrices of the different semantic features and their respective voting options. In this context, soft voting stands for a majority-style election and hard voting indicates the unweighted average of the individual probabilities. Both hard and soft voting are ensemble of (**a**–**c**).

**Figure 16 sensors-22-06063-f016:**
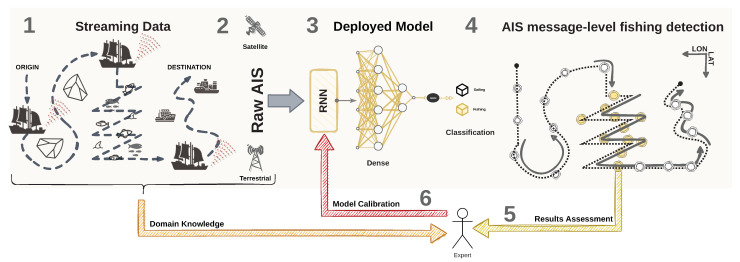
Methodological workflow for the user-in-the-middle fishing detection task. In the diagram, (1) the data are streamed from the vessels’ AIS transceiver; (2) terrestrial or satellite receivers capture the AIS messages; (3) the AIS messages are fed in raw and in sequence to the RNN-based model; (4) the model provides the decision on the vessel activity; (5) the expert verifies the output of the model; (6) the model is updated with the refined labels, and the process is repeated from the first step.

**Table 1 sensors-22-06063-t001:** Benchmark results of our architecture subject to different *window*-and *hidden-layers* sizes. In the column of features, Omeans message-based, T stands for time-based, and D indicates distance-based features.

		Sailing				Fishing				Macro Average					
***w*-Size**	***h*-Size**	**Precision**	**Recall**	**F-Score**	**Support**	**Precision**	**Recall**	**F-Score**	**Support**	**Precision**	**Recall**	**F-Score**	**Support**	**Parameters**	**Feature**
5	32	87.90%	97.87%	92.62%	18,114	79.48%	37.98%	51.40%	3936	83.69%	67.93%	72.01%	22,050	**5704**	O
5	64	91.27%	97.76%	94.40%	18,114	84.67%	56.96%	68.10%	3936	87.97%	77.36%	81.25%	22,050	21,640	O
5	128	92.73%	97.42%	95.01%	18,114	84.50%	64.84%	73.38%	3936	88.62%	81.13%	84.19%	22,050	84,232	O
10	32	93.37%	97.57%	95.42%	13,319	87.92%	71.87%	79.09%	3281	90.64%	84.72%	87.25%	16,600	5864	O
10	64	93.46%	97.60%	95.49%	13,319	88.14%	72.26%	79.42%	3281	90.80%	84.93%	87.45%	16,600	21,960	O
10	128	**93.64%**	97.62%	**95.59%**	13,319	88.32%	73.09%	79.99%	3281	**90.98%**	**85.35%**	**87.79%**	16,600	84872	O
15	32	92.16%	97.10%	94.56%	8941	**88.38%**	72.72%	79.79%	2709	90.27%	84.91%	87.18%	11,650	6024	O
15	64	91.85%	97.10%	94.41%	8941	88.22%	71.58%	79.03%	2709	90.04%	84.34%	86.72%	11,650	22,280	O
15	128	92.41%	96.96%	94.63%	8941	88.01%	**73.72%**	**80.23%**	2709	90.21%	85.34%	87.43%	11,650	85,512	O
5	32	91.95%	95.95%	93.91%	18,391	73.97%	57.78%	64.88%	3659	82.96%	76.86%	79.39%	22,050	**5704**	T
5	64	88.33%	97.06%	92.49%	18,391	70.63%	35.56%	47.30%	3659	79.48%	66.31%	69.89%	22,050	21,640	T
5	128	92.95%	96.33%	94.61%	18,391	77.42%	63.27%	69.63%	3659	85.19%	79.80%	82.12%	22,050	84,232	T
10	32	92.32%	95.40%	93.83%	13,432	77.28%	66.35%	71.40%	3,168	84.80%	80.88%	82.62%	16,600	5864	T
10	64	92.18%	95.81%	93.96%	13,432	78.67%	65.53%	71.50%	3168	85.42%	80.67%	82.73%	16,600	21,960	T
10	128	92.00%	96.02%	93.97%	13,432	79.31%	64.61%	71.21%	3168	85.66%	80.32%	82.59%	16,600	84,872	T
15	32	89.80%	95.31%	92.47%	8947	80.50%	64.15%	71.40%	2703	85.15%	79.73%	81.94%	11,650	6024	T
15	64	90.27%	95.55%	92.83%	8947	81.73%	65.89%	72.96%	2703	86.00%	80.72%	82.90%	11,650	22,280	T
15	128	90.20%	95.76%	92.90%	8947	82.38%	65.56%	73.01%	2703	86.29%	80.66%	82.96%	11,650	85512	T
5	32	86.53%	97.71%	91.78%	18,114	74.01%	30.03%	42.73%	3936	80.27%	63.87%	67.25%	22,050	**5704**	D
5	64	88.15%	97.75%	92.70%	18,114	79.26%	39.51%	52.73%	3936	83.70%	68.63%	72.72%	22,050	21,640	D
5	128	87.45%	**98.10%**	92.47%	18,114	80.06%	35.19%	48.89%	3936	83.75%	66.64%	70.68%	22050	84,232	D
10	32	88.43%	97.28%	92.65%	13,319	81.42%	48.34%	60.66%	3281	84.92%	72.81%	76.65%	16,600	5864	D
10	64	88.72%	97.51%	92.90%	13,319	83.07%	49.65%	62.15%	3281	85.89%	73.58%	77.53%	16,600	21,960	D
10	128	86.64%	96.64%	91.36%	13,319	74.31%	39.50%	51.58%	3281	80.48%	68.07%	71.47%	16,600	84,872	D
15	32	86.14%	95.57%	90.61%	8941	77.11%	49.24%	60.10%	2709	81.62%	72.41%	75.36%	11,650	6024	D
15	64	87.48%	96.39%	91.72%	8941	82.05%	54.49%	65.48%	2709	84.76%	75.44%	78.60%	11,650	22,280	D
15	128	82.83%	96.51%	89.15%	8941	74.68%	33.96%	46.69%	2709	78.75%	65.24%	67.92%	11,650	85,512	D

**Table 2 sensors-22-06063-t002:** Benchmark results of alternative architecture solutions using either GRUs or LSTMs. In the column of features, O means message-based, T stands for time-based, and D indicates distance-based features.

	Sailing				Fishing				Macro Average					
**Unit**	**Precision**	**Recall**	**F-Score**	**Support**	**Precision**	**Recall**	**F-Score**	**Support**	**Precision**	**Recall**	**F-Score**	**Support**	**Parameters**	**Feature**
GRU	93.26%	**97.89%**	**95.52%**	13,319	**89.27%**	71.29%	**79.27%**	3281	**91.27%**	84.59%	**87.40%**	16,600	64,200	O
LSTM	**93.32%**	97.57%	95.40%	13,319	87.92%	**71.62%**	78.94%	3281	90.62%	**84.60%**	87.17%	16,600	85,320	O
GRU	92.21%	95.82%	93.98%	13,432	78.74%	65.69%	71.62%	3168	85.47%	80.75%	82.80%	16,600	64,200	T
LSTM	92.44%	95.76%	94.07%	13,432	78.81%	66.79%	72.30%	3168	85.62%	81.28%	83.19%	16,600	85,320	T
GRU	89.96%	97.31%	93.49%	13,319	83.67%	55.90%	67.02%	3281	86.81%	76.60%	80.26%	16,600	64,200	D
LSTM	89.27%	97.29%	93.11%	13,319	82.69%	52.54%	64.26%	3281	85.98%	74.92%	78.68%	16,600	85,320	D

## Data Availability

The dataset we used is available at https://marinecadastre.gov/AIS accessed 9 August 2022. The source code and additional scripts are included in https://github.com/marthadais/AISclassification accessed 9 August 2022.

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
