# Peer review of "A Semi-Supervised Methodology for Fishing Activity Detection Using the Geometry behind the Trajectory of Multiple Vessels"

_sensors, 2022, doi:10.3390/s22166063_

Round 1
Reviewer 1 Report
General evaluation:
This paper proposes a geometric-driven semi-supervised approach for fishing activity detection and mobility pattern based on Automatic Identification System (AIS) data, by extracting the features describing the geometry of the vessel routes with the use of Recurrent Neural Network (RNN) architectures. In general, the paper is interesting, well-written and fit for the Sensors journal. I think the paper could be accepted after a minor revision. Some comments or suggestions are given to the authors.
Comments:
§ The paper title is better not to contain the abbreviation “AIS”. The title could be modified to be more attractive.
§ There are too much abbreviations such as AIS, SOG, COG, RNN, IUU, UNCLOS, etc. So, it is better to give a list of abbreviations of all jargons before the Introduction section.
§ Regarding the literature review, please consider to cite more relevant papers recently published in Sensors or other MDPI journals, if existing.
§ On Line 49, change “Many efforts were spent searching” to “Many efforts were spent on searching”
§ On Lines 141-150, change “step x in Figure 3” to “Step x in Figure 3”; change “steps x and x” to “Steps x and x”, using the capital letter (“S”) to indicate a special item.
§ The titles of Tables 1 and 2 are too long to follow up. Thus, it is suggested to shorten them and move some explanations to the end of Tables or to the main text.
§ On Line 604, change “Ecological indicators” to “Ecological Indicators”.
§ On Line 636, need to give the university name and country of this PhD thesis in [44].
Author Response
Dear Associate Editor,
Sensors
We thank you and the reviewers for their time and effort in reading and
commenting on our manuscript. We also thank you for the opportunity
to improve this manuscript. Enclosed you will find a new version of our
manuscript entitled “A semi-supervised methodology for fishing activity detection using the geometry behind the trajectory of multiple vessels” and a
detailed response to reviewers.
Kind regards,
The Authors
————————————
Reviewer 1:
1) “The paper title is better not to contain the abbreviation “AIS”. The title
could be modified to be more attractive.”
Answer: Thank you for your suggestion. We modified the title of the
manuscript to make it more attractive as follow: “A semi-supervised methodology for fishing activity detection using the geometry behind the trajectory of multiple vessels”
2) “There are too much abbreviations such as AIS, SOG, COG, RNN, IUU,
UNCLOS, etc. So, it is better to give a list of abbreviations of all jargons
before the Introduction section.”
Answer: Thank you for the suggestion. We included the list of abbreviations
after the conclusion section as indicated in the MDPI template.
3) “Regarding the literature review, please consider to cite more relevant papers
recently published in Sensors or other MDPI journals, if existing.”
Answer: Thank you for your comment. We included 3 more papers from MDPI journals related to our manuscript.
4) “The titles of Tables 1 and 2 are too long to follow up. Thus, it is suggested
to shorten them and move some explanations to the end of Tables or to
the main text.”
Answer: Thank you for your comment. We reduced the title of both tables
and include more information in the main text.
5) “ On Line 49, change “Many efforts were spent searching” to “Many efforts
were spent on searching”
On Lines 141-150, change “step x in Figure 3” to “Step x in Figure 3”;
change “steps x and x” to “Steps x and x”, using the capital letter (“S”)
to indicate a special item.
On Line 604, change “Ecological indicators” to “Ecological Indicators”.
On Line 636, need to give the university name and country of this PhD
thesis in [44]. ”
Answer: Thank you for highlighting these errors. We fixed all and also
checked and revised the whole manuscript.

Reviewer 2 Report
The present work treats about a semi-supervised geometric-driven methodology for supervised fishing activity detection on multi-source AIS tracking messages. The manuscript is interesting and well written. I recommend to publish it subjected to the following modifications:
- Line 2, change “we show” to “it is shown”. Idem in line 7.
- Line 29, change “[6,7]., which” to “[6,7], which”.
- In the figure caption of figure 2 put (a) and (b) in the first line and the explanation of each figure in the line below (a) and (b).
- Line 134, change “[0,360[“ to “[0,360]”.
- In the figure caption of Fig. 7, change “An illustration” to “Illustration”. Idem in Figs. 8, 9, 10, 11, 12, 13, and 14.
Author Response
Dear Associate Editor,
Sensors
We thank you and the reviewers for their time and effort in reading and commenting on our manuscript. We also thank you for the opportunity to improve this manuscript. Enclosed you will find a new version of our manuscript entitled “A semi-supervised methodology for fishing activity detection using the geometry behind the trajectory of multiple vessels” and a detailed response to reviewers.
Kind regards,
The Authors
————————————
Reviewer 2:
1) “ Line 2, change “we show” to “it is shown”. Idem in line 7.
Line 29, change “[6,7]., which” to “[6,7], which”.
In the figure caption of figure 2 put (a) and (b) in the first line and the
explanation of each figure in the line below (a) and (b).
Line 134, change “[0,360[“ to “[0,360]”.
In the figure caption of Fig. 7, change “An illustration” to “Illustration”.
Idem in Figs. 8, 9, 10, 11, 12, 13, and 14. ”
Answer: Thank you for highlighting these errors. We reviewed the manuscript
thoroughly and incorporated the suggestions.
